# The HINTS examination and STANDING algorithm in acute vestibular syndrome: A systematic review and meta-analysis involving frontline point-of-care emergency physicians

Millie Nakatsuka[1,2,3]*, Emma E. Molloy[3]

**1** Discipline of Clinical Ophthalmology and Eye Health, The University of Sydney, Sydney, Australia, **2** Save Sight Institute, The Sydney Eye Hospital, Sydney, Australia, **3** West Moreton Health Service, Ipswich, Queensland, Australia

* Millie.Nakatsuka@health.qld.gov.au

**Data Availability Statement:** The minimal dataset is now available online from https://doi.org/10.25910/tqv0-qz64.

## Abstract

This systematic review aims to evaluate whether point-of-care emergency physicians, without special equipment, can perform the HINTS examination or STANDING algorithm to differentiate between central and non-central vertigo in acute vestibular syndrome with diagnostic accuracy and reliability comparable to more specialized physicians (neuro-ophthalmologists and neuro-otologists). Previous research has concluded that emergency physicians are unable to utilize the HINTS examination with sufficient accuracy, without providing any appropriate education or training. A comprehensive systematic search was performed using MEDLINE, Embase, the Cochrane CENTRAL register of controlled trials, Web of Science Core Collection, Scopus, Google Scholar, the World Health Organization International Clinical Trials Registry Platform, and conference programs and abstracts from six medical organizations. Of the 1,757 results, only 21 were eligible for full-text screening. Two further studies were identified by a manual search of references and an electronic search for any missed studies associated with the authors. Five studies were included in the qualitative synthesis. For the STANDING algorithm, there were two studies of 450 patients who were examined by 11 emergency physicians. Our meta-analysis showed that emergency physicians who had received prior education and training were able to utilize the STANDING algorithm with a sensitivity of 0.96 (95% confidence interval: 0.87–1.00) and a specificity of 0.88 (0.85–0.91). No data was available for the HINTS examination. When emergency physicians are educated and trained, they can use the STANDING algorithm with confidence. There is a lack of evidence regarding the HINTS examination; however, two ongoing studies seek to remedy this deficit.

## Introduction

AVS (acute vestibular syndrome) is a rapid-onset, persistent vertigo or dizziness associated with nausea or vomiting, head-motion intolerance, gait instability, and spontaneous

**Funding:** The author(s) received no specific funding for this work.

**Competing interests:** The authors have declared that no competing interests exist.

nystagmus [1, 2]. It results from an "acute unilateral central or peripheral vestibular lesion that causes a sudden asymmetry of the normal vestibular nuclei neuronal firing rate" [3].

The HINTS examination (head impulse, nystagmus, test of skew) is used to identify central lesions in AVS and distinguish them from peripheral, relatively benign, and more common diagnostic alternatives [2]. This three-step bedside test has been rapidly adopted because it has greater sensitivity (97%) and specificity (99%) than early diffusion-weighted magnetic resonance imaging (MRI) in stroke diagnosis [2].

Further research led to the development of more than five variants to the original 2009 HINTS methodology, as follows: (1) the 2013 HINTS-PLUS, which added a test of acute hearing loss [4]; (2) the 2014 STANDING algorithm (spontaneous nystagmus, direction, head-**i**mpulse test, and standing), which has a modified, four-step test [5]; (3) the 2015 TiTrATE paradigm (timing, triggers, and targeted bedside eye examinations) [6]; and (4) the 2017 ATTEST approach (associated symptoms, timing and triggers, examination signs and testing) [7].

Video-oculography (VOG) or "eye ECG" is used in a further variant, VOG-HINTS [8], which is able to capture subtle oculomotor disturbances [9]. Although VOG-guided care may significantly lower costs, increase health utility, and may result in less missed strokes and improved patient outcomes [10], it has not been widely used due to issues including disruptive eye movements and a high artifact rate [11].

A retrospective review by Rau et al. (2020) identified that only 18% of patients presenting with acute vertigo had a documented HINTS examination [12]. Other prominent studies, including a 2020 systematic review by Ohle et al. [13] and an ongoing phase two trial by Newman-Toker et al. [14], have concluded that only specialized physicians (neuro-ophthalmologists and neuro-otologists) are able to utilize HINTS because its component tests are unfamiliar to most emergency physicians [15]. These studies suggest VOG-assisted diagnosis.

Other studies have instead recommended further investigations. In 2017, Dumitrascu et al. called for the development of a dose-response curve for educational interventions for emergency physicians [16]. In 2020, Hunter proposed that emergency physicians may be able to utilize the HINTS examination with formal education and training [17]. In 2020, Ceccofiglio et al. raised concerns about the practicality of widespread VOG-assisted care because of cost considerations [18].

Our objective was to evaluate whether frontline point-of-care emergency physicians, without specialist equipment, can diagnose central vertigo in AVS using the HINTS examination (or its variants) with sufficient diagnostic accuracy and reliability. We did not intend to conduct a comparative analysis.

## Materials and methods

This systematic review and meta-analysis adheres to the PRISMA statement [19] and is registered on PROSPERO [20].

### Sources and search strategy

In May 2020, we searched MEDLINE, Embase, the Cochrane CENTRAL register of controlled trials, Web of Science Core Collection, Scopus, Google Scholar, and the World Health Organization International Clinical Trials Registry Platform. We did not place limits on document type, study design, language, or publication status. We searched for publications from 2009 onwards (the year HINTS was first introduced). Four assistants translated foreign studies.

In July 2020, we supplemented the above searches by contacting the following organizations and searching programs and abstracts of their annual or biennial conferences: the Neuro-

Ophthalmology Society of Australia, the Neuro-Otology Society of Australia, the Australasian College for Emergency Medicine, the Australian and New Zealand Association of Neurologists, the Royal Australian and the New Zealand College of Ophthalmologists. In August 2020, it was apparent that a wider search (beyond Australia and New Zealand) would be redundant, for the most part, given most international organizations are affiliated with official journals that publish their conference programs, including abstracts for oral and poster presentations. We contacted the International Federation for Emergency Medicine directly to manually search their conference programs and abstracts.

After full-text screening of the studies identified, we further searched every article cited in those studies or associated with their authors. We also contacted the authors of a 2019 poster presentation (Rayner et al. [21]) that intended to provide emergency physicians and trainees with education and training in HINTS.

If we required clarification of details or unpublished data from a study, we initially tried to contact the corresponding author. If no response was received within seven days, we tried again and also searched for alternative contact details using their associated institutions, ResearchGate, Open Researcher and Contributor Identification, and social media (Facebook and Twitter). If we received no response within another week, we contacted the subordinate authors. We excluded studies that provided no contact information. We excluded those whose contacts provided no response within a one month period. When the contact was known to be on leave, we attempted contact after their return.

## Study selection criteria

After training and pilot testing in accordance with best practice guidelines for abstract screening [22], two independent reviewers (MN and EM) conducted screening in three stages using pre-determined criteria. All disagreements were resolved in subsequent discussion, without the independent third reviewer.

**Stage 1 screening.** One reviewer (MN) excluded irrelevant studies including book or book sections, studies based on animal subjects or models, studies involving people younger than 18 years old, and duplicates.

**Stage 2 screening.** Two independent reviewers (MN and EM) screened titles and abstracts. The inclusion criteria were: (1) primary research with original data; (2) patients presented to emergency settings with AVS (vertigo or dizziness) as primary complaint; and (3) one of the outcomes of interest was the diagnostic accuracy the HINTS examination (or any of its three individual components) or its variants. This third criterion was not mandatory, as we wanted to minimize the likelihood of missing diagnostic accuracy data presented within the full-text or only known only by the authors. Therefore, for each study, we used four questions to estimate the likelihood of it providing the required data at the required level of detail. In order for a study to progress to the next stage of screening, it required a minimum score of four points out of ten. We excluded studies of fewer than five research subjects and those that utilized VOG-associated technology or telemedicine.

**Stage 3 screening.** Two independent reviewers (MN and EM) reviewed the full-text of each selected article. Our inclusion criteria were: (1) patients were examined by emergency physicians or trainees; (2) AVS (new-onset vertigo or dizziness) was the primary presenting complaint; (3) measures of diagnostic accuracy were available or data were available from which these could be derived; and (4) the final diagnosis was determined by at least one qualified physician, based on all available clinical information.

We excluded studies: (1) that included patients with a history of chronic or transient AVS; (2) that included patients who initially presented with gross neurological deficits, such as

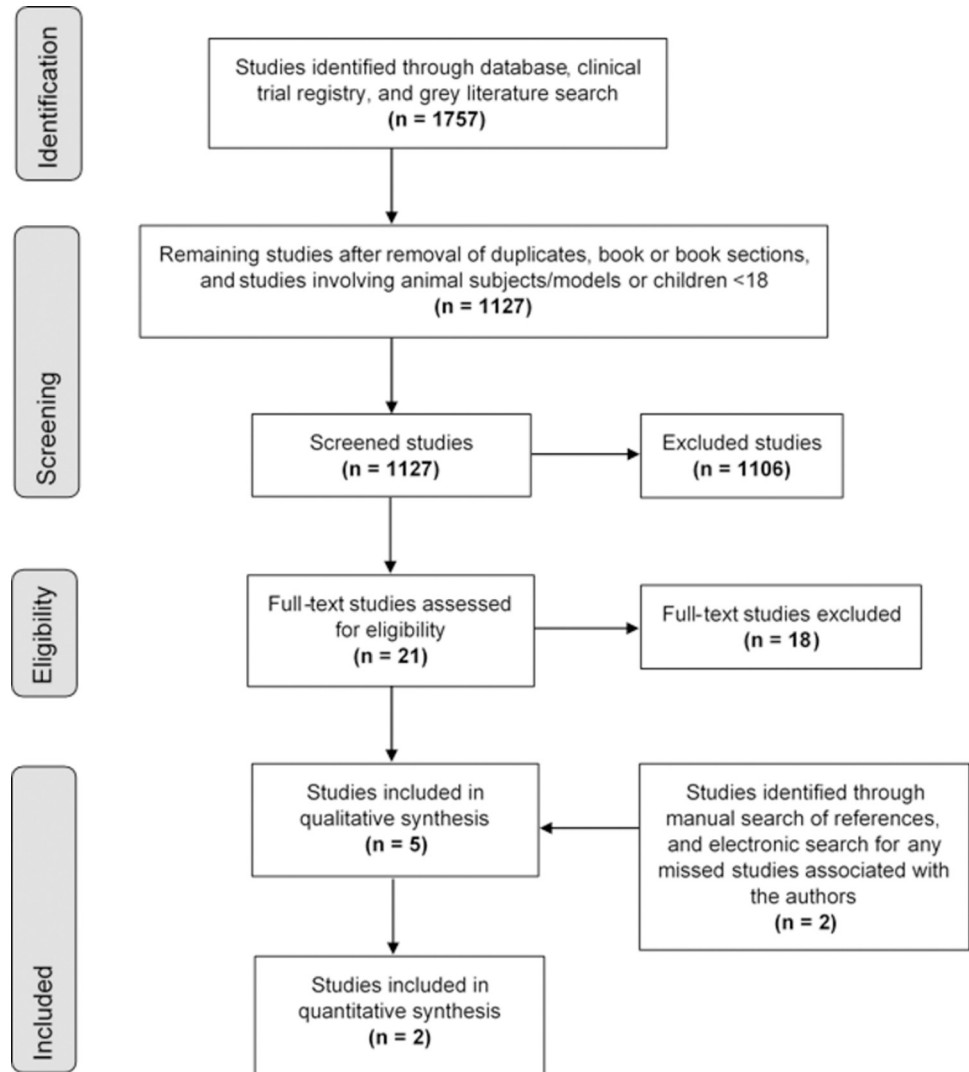

**Fig 1. Overview of study selection.** The diagram summarizes search results between January 1, 2009, and May 14, 2020. MEDLINE found 326 records, Embase 723 records, Cochrane Central 20 records, Web of Science Core Collection 131 records, Scopus 169 records, Google Scholar 200 records, and International Clinical Trials Registry Platform (World Health Organisation) 23 records. The search was updated during July and August 2020 by hand searching gray literature, identifying 165 records. All 1127 records were screened for eligibility, with 2 new studies meeting the inclusion criteria identified in January 2021.

hemi-paresis; (3) where AVS was deemed to result from a general medical disorder, such as orthostatic hypotension [23], or from obvious neurological disorders other than stroke, such as traumatic brain injuries [23]; and (4) where AVS was deemed to result from similar rare causes. We also excluded studies that were concerned only with AVS in peripheral vestibular disorders and isolated stroke syndromes. Fig 1 provides an overview of the study selection.

## Data collection

We developed data extraction tables that were refined after a pilot test using a randomly selected study. One reviewer (MN) extracted data, which was cross-checked by the second reviewer (EM). For one particular study [5], we additionally reviewed the duplicate 2015 study

[24], to screen for inconsistencies. All disagreements were resolved in subsequent discussions between the first and second reviewer, without involving the third reviewer.

### Risk of bias in individual studies

Two independent reviewers (MN and EM) evaluated the quality of each study against the 11 recommended quality items derived from QUADAS-2 tool [25]. We added an additional domain to evaluate the suitability of the education and training program for emergency physicians: "Did the operators of the index test(s) receive an appropriate level of education and training?" Disputes were again resolved by discussion. We used the *robviz* package (https://github.com/mcguinlu/robvis) with R software (R Foundation for Statistical Computing, Vienna, Austria) to summarize the overall results [26].

### Statistical analysis

We analyzed the data on MetaDiSc 1.40 (http://www.hrc.es/investigacion/metadisc_en.htm). We adopted a random effects model and performed the meta-analysis using the Der Simonian and Laird procedure to calculate the point estimate of pooled sensitivity and specificity, with a 95% confidence interval [27]. Because one data point from Vanni et al. (2014) was zero, we added 0.5 to all values [5].

### Risk of bias across studies

We used Cochran's Q test, the chi-square with *p*-values, and the $I^2$ heterogeneity statistic to estimate whether variation between two studies was beyond that reasonably expected by chance. We note that, with only two studies, these tests have limited value and could be misleading. Therefore, they did not influence us to conduct a meta-analysis; we relied on our assessments of the risk of bias in individual studies.

We were unable to obtain the data required to assess interobserver variability, nor was it possible to compare the Cohen's kappas in Vanni et al. (2014) to the Fleiss kappas in Vanni et al. (2017) [5, 28].

## Results

### Study selection

Our initial searches across five databases and two registries of clinical trials identified 1410 articles. We subsequently reviewed conference programs and abstracts of scientific meetings from six medical organizations, resulting in a further 165 works, including recordings of conference presentations. After stage one screening 1127 studies remained. Of these, 1106 were excluded after reviewing titles and abstracts. We then scrutinized the full-text of the remaining 21 studies; only three fulfilled the selection criteria. We noted that Vanni et al. (2015) was as a duplicate of Vanni et al. (2014) [5, 24]. A supplemental search identified one further study, Ceccofiglio et al. (2020) [18]. We further included a study proposal by Rayner et al., from which we hoped to obtain relevant data before finishing our analysis [21]. Thus, a total of five studies were included in the qualitative synthesis.

As of January 15th 2021, the Gerlier et al. study is ongoing, while the Rayner et al. study is still in its protocol stage [21, 29].

### Study characteristics

All five studies were single-center studies in which an appropriate level of education and practical training was provided (or was intended to be provided) to emergency physicians. Three

**Table 1. Patient and physician characteristics in the studies included in the qualitative synthesis.**

| | Vanni 2014 | Vanni 2017 | Ceccofiglio 2020 | Gerlier | Rayner |
|---|---|---|---|---|---|
| References | 5 | 28 | 18 | 29 | 21 |
| **Patient characteristics** | | | | | |
| Total number | 98 | 352 | 24 | 232 | - |
| Age (years) | $\bar{x} = 60 \pm 16.3$ | $\bar{x} = 58 \pm 18$ | $\bar{x} = 54$ | - | - |
| Male | n = 42 (42.9%) | n = 142 (40.3%) | n = 6 (25.0%) | - | - |
| No nystagmus | n = 13 (14.3%) | n = 76 (21.5%) | - | n = 0 (0.00%) | - |
| CT | n = 31 (31.6%) | n = 137 (38.9%) | - | - | - |
| MRI | n = 10 (10.2%) | n = 27 (7.7%) | - | n = 232 (100%) | - |
| Central vertigo | n = 11 (11.2%) | n = 40 (11.4%) | n = 0 (0.00%) | - | - |
| **Participating physicians** | | | | | |
| Physicians | n = 5 | n = 6 | n ≤ 40 (?) | n = 8 | n = 15 |
| Speciality | Emergency medicine | Emergency medicine | Emergency medicine | Emergency medicine | Emergency medicine |
| Examiner | Independent physician | Independent physician | Treating physician | Independent physician | Unknown |
| **Quality and quantity of physician education and training** | | | | | |
| Participation | n = 5 (100%) | n = 6 (100%) | n = 5 (?%) | n = 8 (100%) | n = 15 (100%) |
| Education | 5 hours of lectures | 4 hours of lectures | 4 hours of lectures | 2 hours of lectures | 1 hour of lecture |
| Practical training | 1 hour of workshop | 8 hours of workshops | 8 hours of workshops | 8 hours of workshops | Yes |
| Supervised placement | No | 4 weeks with neuro-otologist | 4 weeks with neuro-otologist | No | Yes |
| Assessment | 15 proctored exams | 10 proctored exams | 10 proctored exams | No | Yes |

The table summarizes the characteristics of the participating patients and emergency physicians. As of January 15th 2021, the Gerlier et al. study is ongoing, while the Rayner et al. study is still in its protocol stage [21, 29].

of the five studies were prospective cohort studies: Vanni et al. (2017), Gerlier et al., and Rayner et al. [21, 28, 29]. Vanni et al. (2014) was a prospective, quasi-randomized, controlled trial [5]. The only retrospective review was that of Ceccofiglio et al. (2020) [18].

Three of the studies were conducted in Italy by a largely unchanged core group of physicians, all using only the STANDING algorithm: Vanni et al. (2014), Vanni et al. (2017), and Ceccofiglio et al. (2020) [5, 28, 18]. By contrast, in the Gerlier et al. study, participating physicians conducted both the STANDING algorithm and the original HINTS examination for each patient [29]. Rayner et al. intend to apply only the original HINTS examination [21]. The characteristics of the participating patients and physicians are summarized in Table 1. The minimal underlying data set is available online [30].

## Risk of bias in individual studies

We analyzed the 12 domains based on adjusted recommended quality items derived from QUADAS-2 tool [25]. The main reasons for downgrading studies were no response bias, no data bias, and lack of representativeness of the population sample. Only two of the studies, Vanni et al. (2014) and Vanni et al. (2017), were of overall quality high enough to be included in the quantitative analysis [5, 28]. The results are summarized in Fig 2.

## Result of individual studies

Three studies provided data for the diagnostic accuracy of the STANDING algorithm. Rayner et al. had no data, while Gerlier et al. did not provide data [21, 29].

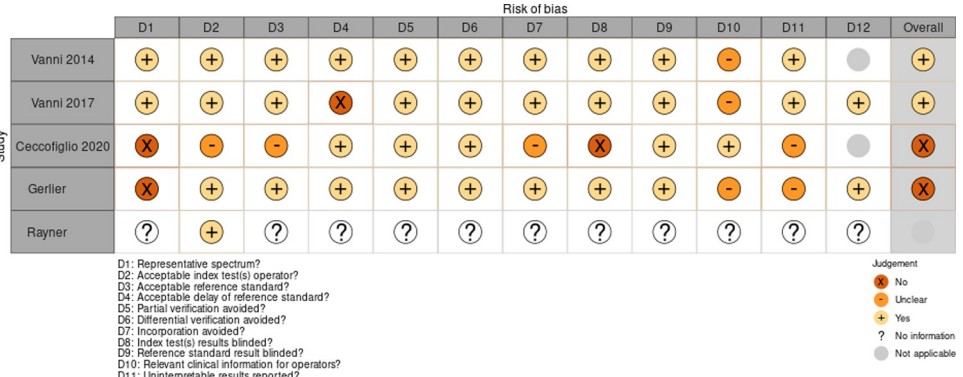

**Fig 2. Traffic light plot of the risk of bias within studies.** The diagram summarises the risk of bias within the individual studies, based on QUADAS-2 tool derived adjusted recommended quality items for each of the twelve domains. QUADAS = Quality Assessment tool for Diagnostic Accuracy Studies.

In the Vanni et al. (2014) study, 92 patients were examined by five educated and trained emergency physicians, with a sensitivity of 0.96 (95% confidence interval: 0.83–0.99), and a specificity of 0.87 (82–0.90) [5].

In the Vanni et al. (2017) study, 352 patients were examined by six educated and trained emergency physicians, with a sensitivity of 1.00 (0.72–1.00), and a specificity of 0.94 (0.82–0.90) [28]. Thirteen patients withdrew from this study; four were lost to follow-up and nine were unable to attend follow-up.

In the Ceccofiglio et al. (2020) study, 24 patients were examined. We were unable to verify the number of emergency physicians who performed these examinations but we were able to determine that five of them had received education and training as participants in the Vanni et al. (2017) study [18, 28]. The specificity was 37.5%. The sensitivity could not be calculated as there were no cases of central vertigo.

## Synthesis of results

Diagnostic data was available from two studies, with 450 patients examined by 11 emergency physicians who were educated and trained to perform and interpret the STANDING algorithm. The sensitivity and specificity of individual studies are summarized in Fig 3, as pooled values with 95% confidence intervals. Pooled sensitivity was 0.96 (0.87–1.00) and pooled specificity was 0.88 (0.85–0.91).

## Risk of bias across studies

Our analysis suggests heterogeneity for specificity (chi-square = 4.49 [$p = 0.0341$]; $I^2 = 77.7\%$) but no heterogeneity for sensitivity (chi-square = 0.99 [$p = 0.3188$]; $I^2 = 0.0\%$).

Patient characteristics are summarized in Table 1. We noted minimal variance between the two studies. Given this sufficient homogeneity, the data can usefully be combined to answer broader questions than could be addressed by the individual studies alone.

## Discussion

Previous studies of HINTS examination use by emergency physicians have differentially supported two general streams of thought. Some have suggested that emergency physicians are unable to use the HINTS examination with sufficient accuracy, and have suggested a move toward VOG-based care (e.g. the ongoing "eye ECG" trial by Newman-Toker et al. [14]). A

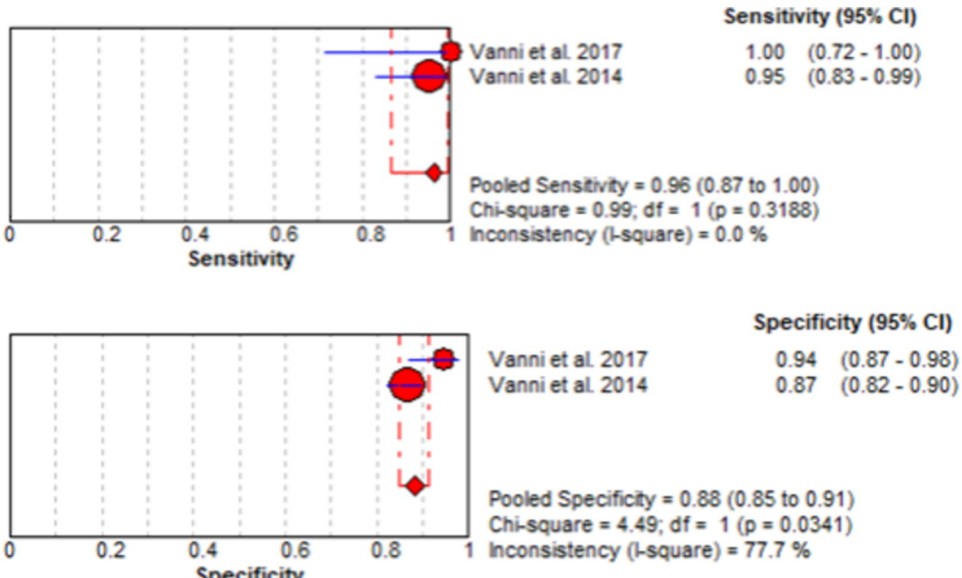

**Fig 3. Meta-analysis of the diagnostic accuracy of the STANDING algorithm in trained emergency physicians.**
The forest plot shows the individual and pooled sensitivity and specificity with 95% confidence intervals. A random-effects model was used. Cochran's Q test, chi-square with $p$ values, and I$^2$ were calculated to estimate whether the variation between two studies was beyond that reasonably expected by chance.

systematic review by Ohle et al. (2020) also supports this view; it reported a sensitivity of 83% and a specificity of 44% [13]. However, this was based solely on the Kerber et al. (2015) study [31], which is limited in that it did not provide any education or training to emergency physicians; the examiners were fellowship-trained in vascular neurology. Moreover, the outcome was purely MRI-based, with a substantial incidence of patient withdrawal (48/320 patients or 15%).

Other studies have not concluded against emergency physicians, but have instead called for further research into education and training required, for example, Hunter (2020), Dumitrascu et al. (2017), and Ceccofiglio et al. (2020) [16–18]. Concerns have been expressed about the feasibility of expensive VOG-HINTS examinations in non-major city hospitals. This avenue of inquiry is particularly important for addressing health inequality.

We designed our study to follow this line of enquiry. We sought to focus on emergency physicians, explicitly measuring the education and training they received, and evaluating their use of the HINTS examination and its four variants. We adopted a deliberately extensive search strategy, without limitations, in an attempt to improve upon Ohle et al. (2020), who had pre-emptively excluded a large volume of potentially relevant studies, including ongoing trials, non-peer reviewed journals, and unpublished data [13].

Our systematic review and meta-analysis thus provides the first quantitative summary estimate of the sensitivity and specificity achieved by educated and trained emergency physicians in performing and interpreting the STANDING algorithm to rule out central pathology in patients with AVS. Our results show that, with education and training, emergency physicians can independently identify central vertigo in AVS using the STANDING algorithm. This was achieved with high diagnostic accuracy, with a pooled sensitivity of 0.96 (0.87–1.00) and a specificity of 0.88 (0.85–0.91). This finding is significant support for the contention that emergency physicians have been prematurely dismissed as effective examiners.

Currently, data for evaluating the use of the original HINTS examination by emergency physicians are not available. However, it is pleasing to note that there is ongoing (Gerlier et al.)

and proposed (Rayner et al.) research on the diagnostic accuracy of the HINTS examination by appropriately educated and trained emergency physicians [21, 29]. Our search found no studies where emergency physicians used the HINTS-PLUS, TiTrATE paradigm, or ATTEST approach.

We wish to highlight that, although "HINTS appears to be accurate independent of the presence or absence of other neurological signs" [32], the original test cannot be conducted in patients who are no longer symptomatic or without spontaneous nystagmus. Furthermore, many neurologists have expressed concern that other neurological signs are undervalued. There is increasing evidence to support that vestibulospinal signs, in particular, are "probably strong predictors of a central cause": for example, a systematic review by Tarnutzer et al. (2011) [33]. Edlow and Newman-Toker (2016) noted that the first of five prudent questions to pose when assessing acute dizziness and vertigo, was regarding the patient's ability to sit or stand without assistance [32]. Carmona et al. (2015) noted that truncal ataxia was "an easy sign, even for physicians without specific training in clinical neurology" [34]. In essence, the STANDING algorithm is a HINTS examination plus positional nystagmus and evaluation of standing position and gait–an assessment that can be performed in all patients.

The ATTEST approach (the revised and renamed TiTrATE paradigm) incorporates aspects of the patient history combined with targeted bedside examinations to best inform a differential diagnosis. Although there have been no clinical trials to date, even among physicians of other specialities, it is considered to be "supported by a very strong evidence base in the speciality literature" [7].

## Limitations

Despite its design being a major strength of our study, it has inherent limitations that contribute to uncertainty and bias, particularly methodological diversity. Capturing as many studies as possible ensured that some of our data came from subsections of studies with an array of different study types, designs, and objectives. Many studies did not record as much data as we required. For example, we were unable to assess interobserver reliability for the STANDING algorithm because the particular physician who examined each patient was not recorded.

Furthermore, our meta-analysis contained only two studies. We were unable to perform an individual patient data meta-analysis (considered the gold standard) because of a lack of individual patient data.

Other limitations included:

1. Three of the five studies were conducted by a common working group of physicians: Vanni et al. (2014), Vanni et al. (2017), and Ceccofiglio et al. (2020) [5, 18, 28]. Limited and repetitive authorship is not ideal and increases the risk of bias, raising concerns that findings may vary significantly in different settings and populations.

2. Vanni et al. (2017) mandated active follow-up with a neuro-otologist at one week and three months [28]. This resulted in the withdrawal of 13 of 365 patients.

3. Lack of response and/or data decreased the amount of gray literature that could be screened, with the majority of contacts repeatedly failing to respond and many records incomplete or missing. A similar pattern of lack of response and missing data also affected the fives studies in our review.

4. We used MetaDiSc for meta-analysis despite its outdated Moses-Littenberg method, because we did not require a hierarchical or bivariate analysis.

## Other considerations

(1) Currently, no studies have looked at the dose-response curve, or quality and quantity of the education and training required for emergency physicians. All of the five studies included in the qualitative synthesis conducted (or planned to conduct) both education and practical training. Gerlier et al. was the only study which did not conduct supervised placements or assessments [29].

(2) Similarly, no studies have looked at the value of subsequent ongoing education and refresher courses to maintain skills and confidence in emergency physicians. Although the level of experience, confidence, and clinical utility will vary significantly between practice sites and individual physicians, operator proficiency is likely to decrease over time. Research on other examinations, such as a 2019 study by Schwid et al. suggests that even brief educational interventions can boost confidence to perform and supervise point-of-care ultrasound [35].

(3) Health inequity is a major problem in many countries. Health resource maldistribution is of particular concern, with governments over the last decade backing centralization, and either merging or closing services in Australia, the United States, Canada, Germany, and France [36].

While VOG-HINTS and telemedicine may appear very attractive, concerns have been expressed about the practicality of widespread VOG-assisted care because of cost considerations, including Ceccofiglio et al. (2020) [18]. For example, in Australia, only 25% of public hospitals are located in major cities [37]. Many rural and remote emergency departments have no facilities to conduct pathology tests or advanced imaging; they rely on emergency retrieval services to transport critical patients (and specimens). The cost of VOG is not feasible in such settings, especially with the strain on resources compounded by the coronavirus pandemic.

Three further significant logistical concerns with VOG and telemedicine are that: A) training may be required to use the equipment; B) equipment may malfunction, or face other technical issues such as a slow internet connection; and C) there is limited availability of on-call city specialists for emergency telemedicine consults, particularly after-hours.

(4) A pleasing new development is the increasing interest in using smartphones to record eye movements. Parker et al. (2021), conducted a preliminary study which supports the concept of smartphone-based applications as an alternative to traditional VOG [38].

(5) There is a large body of literature concerned with the loss of clinical skills and growing dependence on technology in medicine [39]. Fear of missing a low probability diagnosis and litigation has also resulted in defensive medicine, increasing the rates of unnecessary diagnostic imaging [40]; this places emergency physicians at particularly high risk due to the nature of their work.

A decade long study by Kanzaria et al. (2015) identified that dizziness-related imaging in emergency settings had increased by 169%, without a corresponding increase in the diagnosis of cerebrovascular disease among these patients [41]. The largest increase of 281% occurred in patients aged 20 to 44, who are relatively at low risk of such events [41]. A recent case presented by Khachatoorian et al. also warns of the potential dangers of misdiagnosis when a thorough history and physical examination are omitted [42].

## Conclusions

The results of our systematic review and meta-analysis are particularly significant for less privileged hospitals that have minimal access to specialists and VOG-assisted technology. We believe that previous studies were unable to demonstrate an acceptable level of diagnostic accuracy because they did not provide prior education and training. Our results show that by addressing the lack of education and training in the STANDING algorithm, frontline

emergency physicians can become equipped to make rapid point-of-care decisions, even in rural or remote settings. We note that two ongoing studies are focusing on providing education and training for the HINTS examination in emergency physicians.

Future research is required to focus on the quality and quantity of the education and training required, as well to assess the value of subsequent ongoing education and refresher courses to preserve operator proficiency.

## Supporting information

**S1 Checklist.**
(DOCX)

## Acknowledgments

I am sincerely grateful to supervisor Dr. Mark Paine (neuro-otology and neuro-ophthalmology specialist), Syeda Azim (statistician), and the team of librarians and translators.

## Author Contributions

**Conceptualization:** Millie Nakatsuka, Emma E. Molloy.

**Data curation:** Millie Nakatsuka, Emma E. Molloy.

**Formal analysis:** Millie Nakatsuka.

**Investigation:** Millie Nakatsuka.

**Methodology:** Millie Nakatsuka, Emma E. Molloy.

**Project administration:** Millie Nakatsuka.

**Resources:** Millie Nakatsuka.

**Validation:** Millie Nakatsuka, Emma E. Molloy.

**Visualization:** Millie Nakatsuka, Emma E. Molloy.

**Writing – original draft:** Millie Nakatsuka, Emma E. Molloy.

**Writing – review & editing:** Millie Nakatsuka, Emma E. Molloy.

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
