## [Decision Letter · Decision Letter 0]

10 Dec 2021

PONE-D-21-34436The HINTS examination and STANDING algorithm in acute vestibular syndrome: a systematic review and meta-analysis involving frontline point-of-care emergency physiciansPLOS ONE

Dear Dr. Nakatsuka,

Thank you for submitting your manuscript to PLOS ONE. After careful consideration, we feel that it has merit but does not fully meet PLOS ONE’s publication criteria as it currently stands. Therefore, we invite you to submit a revised version of the manuscript that addresses the points raised during the review process.

We look forward to receiving your revised manuscript.

Kind regards,

Diego Kaski, PhD MBBS

Academic Editor

PLOS ONE

Journal Requirements:

Reviewers' comments:

Reviewer's Responses to Questions

**Comments to the Author**

1. Is the manuscript technically sound, and do the data support the conclusions?

Reviewer #1: Yes

Reviewer #2: Partly

2. Has the statistical analysis been performed appropriately and rigorously? 

Reviewer #1: Yes

Reviewer #2: Yes

3. Have the authors made all data underlying the findings in their manuscript fully available?

Reviewer #1: Yes

Reviewer #2: Yes

4. Is the manuscript presented in an intelligible fashion and written in standard English?

Reviewer #1: Yes

Reviewer #2: Yes

5. Review Comments to the Author

Reviewer #1: The authors conducted a systematic review to evaluate whether point of care emergency physicians without special equipment can perform the HINTS plus examination or the STANDING algorithm to differentiate peripheral versus central causes of vertigo.

They found 21 eligible studies. The STANDING algorithm form Florence, Italy, included 450 patients examined by 11 ER HINTS-trained ED physicians, These physician not only achieved a sensitivity of 096 (CI 087-1.00) and specificity of 0.88 (CI 0.85 -0.91) and an excellent Cohen’ kappas with subsequent Neurotology evaluation. The initial HINTS examination performed by a neurophthalmologist/neurotologist, involved ED patients and transferred patients from other referring hospital ED without a diagnosis.

One paper studied the effect of training neurologists and ED physicians regarding the HINTS performance, and this training had a short-term impact on correct diagnosis/treatment and decreased number of unnecessary imaging. One study also raised concerns about the cost of VOG-assisted care because of price; this however, will be well balanced by the fact that 1. A permanent record of the findings will be valuable in follow up 2. The cost substantially less that the cost of unnecessary imaging, admission, etc.

I believe that the authors did an excellent review of what is available in the literature. The authors are correct on Vanni’s, et al pioneering training of a small number of ED physicians, who evaluated a large number of patients in time. I believe that this is important, is not only the initial training but also sustained exposure that leads to expertise.

Minor points to address

In the discussion section, this reviewer believes that the authors can state the opinion of Vanni, et al who provided their research team of ED physicians with proper training, and found that they perform the STANDING protocol effectively." Not clearly stated, however, is the need for sustained exposure" If they see only one case a week, confidence will remain weak . An increasing number of ED physicians who focus their attention to vertigo/dizziness, thus increasing enthusiasm is noticeable.

I also think that the diagnosis may not be straightforward with the first clinical exam. In such cases, Video recording may be the best choice. The cost of video-oculography is not very expensive, and it provides a record of the initial findings, which may become critical in the short term for management. Let us draw an analogy with an Echo obtained in a patient with a heart murmur (the goal is precision). If you agree may want to add.

Video- oculography of course would no be needed for BPPV. It is true that local community hospital will need to rely on Telemedicine or refer the patient to a larger facility. It is possible though that common diagnosis such as BPPV or vestibular neuritis could be identify easily by the local ED physician. (If you agree may want to add)

Finally, increase literacy among all future physicians pertaining Neurology and Vestibular Medicine is also predictable in the not so far future. At our School, the curriculum is quite ambitious, and students as they advanced in heir training and early practice will become more aware that this is expected from them. (In the spirit of Education may want to add)

Reviewer #2: In the Stage 1 Screenig the authors state that the excluded irrellevant studies but ignore a lot of papers which use vestibulospinal signs in AVS

"imbalance" or more properly truncal ataxia deserves a special paragraph, especially if quantify (Lee H, Neurology (2006) 67: 1178-83 – Carmona et al doi: 10.3389/fneur.2016 – Vanni et al doi: 10.3389/fneur.2017)

a- truncal ataxia is an specific manifestation of the vestibulospinal compromise and its value in the physical examination is the same of the ocular findings

b- At least in the both side of the spectrum: No ataxia never central, ataxia grade III (unable to walk) is always central (Kiersten L. Gurley, MD1, 2,3 Jonathan A. Edlow, MD, Semin Neurol 2019;39:27–40 - Jonathan A. Edlow, MD, Kiersten L. Gurley, MD,and David E. Newman-Toker https://doi.org/10.1016/j.jemermed.2017.12.024 -Diagnosing Stroke in Acute Dizziness and Vertigo Pitfalls and Pearls Ali S. Saber Tehrani, MD; Jorge C. Kattah, MD; Kevin A. Kerber, MD, MS; Daniel R. Gold, DO; David S. Zee, MD; Victor C. Urrutia, MD; David E. Newman-Toker, MD, PhD Stroke, March 2018)

Other vestibulospinal signs: Babinski asinergia sign, falling from sitting with closed eyes and arm over the chest can help (A New Diagnostic Approach to the Adult Patient with Acute Dizziness. Edlow JA, Gurley KL, Newman-Toker DE J Emerg Med. 2018 Apr; 54(4):469-483. doi: 10.1016/j.jemermed.2017.12.024. Ideally, have the patient walk unassisted, but for severely nauseated patients too symptomatic to walk, test for truncal ataxia by asking the patient to sit upright in the stretcher with arms crossed. Patients who cannot walk or sit up unassisted are unsafe for discharge and are more likely to have a stroke (or other CNS pathology) rather than vestibular neuritis (27, 44, and 77). Although American emergency physicians are uncomfortable using HINTS testing and instead overuse CT, one study reported that specially trained emergency physicians using these bedside examination elements decreased both CT use and hospitalization

You compare HINTS against STANDING, but HINTS is about AVS but STANDING is about Differential Diagnosis of Vertigo in the Emergency Department, it includes for example, another etiologies of vertigo like BPPV so a comparison is not posible

6. PLOS authors have the option to publish the peer review history of their article (what does this mean?). If published, this will include your full peer review and any attached files.

Reviewer #1: No

Reviewer #2: **Yes: **Sergio Carmona

---

## [Author Response · Author response to Decision Letter 0]

31 Jan 2022

Thank you for the opportunity to submit a revised draft of our manuscript “The HINTS examination and STANDING algorithm in acute vestibular syndrome: a systematic review and meta-analysis involving frontline point-of-care emergency physicians” for publication as a Research Article in PLOS ONE. We appreciate the time and effort that you and the reviewers have dedicated to providing feedback. We are grateful for the insightful comments on and valuable improvements to our manuscript. 

We have incorporated most of the suggestions made by the reviewers. Below, we provide point-by-point responses to the reviewers’ comments. All page numbers refer to the revised manuscript file (with tracked changes). 

The minimal underlying dataset has been submitted to The University of Sydney eScholarship Repositary. We will contact you as soon as our dataset is online to revise refence number 30 (Page 23). 

We have approved the revised manuscript and agree with its submission to PLOS ONE. There are no conflicts of interest to declare.

We look forward to hearing from you at your earliest convenience.

---

## [Decision Letter · Decision Letter 1]

17 Mar 2022

The HINTS examination and STANDING algorithm in acute vestibular syndrome: a systematic review and meta-analysis involving frontline point-of-care emergency physicians

PONE-D-21-34436R1

Dear Dr. Nakatsuka,

We’re pleased to inform you that your manuscript has been judged scientifically suitable for publication and will be formally accepted for publication once it meets all outstanding technical requirements.

Kind regards,

Diego Kaski, PhD MBBS

Academic Editor

PLOS ONE

Additional Editor Comments (optional):

Reviewers' comments:

Reviewer's Responses to Questions

**Comments to the Author**

1. If the authors have adequately addressed your comments raised in a previous round of review and you feel that this manuscript is now acceptable for publication, you may indicate that here to bypass the “Comments to the Author” section, enter your conflict of interest statement in the “Confidential to Editor” section, and submit your "Accept" recommendation.

Reviewer #1: All comments have been addressed

Reviewer #2: All comments have been addressed

2. Is the manuscript technically sound, and do the data support the conclusions?

Reviewer #1: Yes

Reviewer #2: Yes

3. Has the statistical analysis been performed appropriately and rigorously? 

Reviewer #1: Yes

Reviewer #2: Yes

4. Have the authors made all data underlying the findings in their manuscript fully available?

Reviewer #1: Yes

Reviewer #2: Yes

5. Is the manuscript presented in an intelligible fashion and written in standard English?

Reviewer #1: Yes

Reviewer #2: Yes

6. Review Comments to the Author

Reviewer #1: The questions answered, suggestions incorporated, good paper.. I believe that the manuscript summarizes the state of the art om the topic. The authors raise concern that the that VOG recordings could negatively impact on a less accurate neurologic examination.

The VOG advantages are multiple. one important one is the ability to quantitate the findings, a second very important feature is to examine the eye movements with total fixation block, which is otherwise impossible. This of course shpuld never replace the complete neurologic exam , including posture and gait when possible.

Reviewer #2: Congrats, a good review which demonstrates that the AVS management is still not known by general physicians in the ER

7. PLOS authors have the option to publish the peer review history of their article (what does this mean?). If published, this will include your full peer review and any attached files.

Reviewer #1: No

Reviewer #2: **Yes: **Sergio Carmona

---

## [Editor Report · Acceptance letter]

11 Apr 2022

PONE-D-21-34436R1 

The HINTS examination and STANDING algorithm in acute vestibular syndrome: a systematic review and meta-analysis involving frontline point-of-care emergency physicians 

Dear Dr. Nakatsuka:

I'm pleased to inform you that your manuscript has been deemed suitable for publication in PLOS ONE. Congratulations! Your manuscript is now with our production department. 

Kind regards, 

on behalf of

Dr. Diego Kaski 

Academic Editor

PLOS ONE